# Development of Environmentally Friendly Wool Shrink-Proof Finishing Technology Based on L-Cysteine/Protease Treatment Solution System

**DOI:** 10.3390/ijms232113553

**Published:** 2022-11-04

**Authors:** Bo Li, Jiaying Li, Yanqin Shen, Hailiang Wu, Yanli Sun, Pengfei Zhang, Meihui Yang

**Affiliations:** 1School of Textile Science and Engineering, Xi’an Polytechnic University, Xi’an 710048, China; 2Key Laboratory of Functional Textile Material and Product (Xi’an Polytechnic University), Ministry of Education, Xi’an 710048, China

**Keywords:** wool, shrink-proof finishing, protease, L-cysteine

## Abstract

The particular scale structure and mechanical properties of wool fiber make its associated fabrics prone to felting, seriously affecting the service life of wool products. Although the existing Chlorine–Hercosett treatment has a remarkable effect, it can lead to environmental pollution. Therefore, it is of great significance to develop an environmentally friendly and effective shrink-proof finishing technology. For this study, L-cysteine was mixed with protease to form a treatment solution system for shrink-proof finishing of wool fibers. The reduction performance of L-cysteine and its effect on wool were compared with those of other reagents, demonstrating that L-cysteine has an obvious reduction and destruction effect on the wool scale layer. Based on this, L-cysteine and protease 16L were mixed in a certain proportion to prepare an L-cysteine/protease treatment solution system (L/PTSS). The shrink-proof finishing of a wool top was carried out by the continuous multiple-padding method, and the processing parameters were optimized using the response surface method. The results indicated that when the concentrations of L-cysteine and protease 16L were 9 g/L and 1 g/L, respectively, the wool was padded five times at 50 °C, and each immersion time was 30 s, the felt ball density of the treated wool reduced from 135.86 kg/m^3^ to 48.65 kg/m^3^. The structure and properties of the treated wool were also characterized using SEM, TG, and tensile strength tests, which indicated that the fiber scale structure was stripped evenly. Meanwhile, the treated fibers still retained adequate thermal and mechanical properties, indicating suitable application value. XPS, FT-IR, Raman, UV absorbance, and other test results revealed the reaction mechanism of L/PTSS with the wool fibers. After L-cysteine rapidly reduced the disulfide bonds in wool, protease can hydrolyze peptide chains more effectively, causing the scale layer to gradually peel off. Compared with the chlorination method and other protease shrink-proof technologies, L/PTSS can achieve the finishing effect on wool rapidly and effectively, without causing excessive pollution to the environment. The conclusions of this study provide a foundation for the development and industrial application of biological enzyme shrink-proof finishing technology.

## 1. Introduction

Wool fiber is widely used in various fields of industrial production and daily life due to its various advantages, including soft luster, elasticity, hygroscopicity, and anti-static nature. These properties are closely related to the structure and performance of wool. However, it is also precisely because of the unique scale structure and mechanical properties of its fibers that wool products are prone to obvious felt shrinkage changes when subjected to external mechanical action in a humid and hot environment [1,2]. Moreover, this felt shrink cannot be recovered, seriously affecting the wearing performance and service life of wool products. Therefore, shrink-proof treatment is an indispensable finishing process in the production of wool fabric. Traditionally, the most commonly used finishing technique in industrial production is Chlorine–Hercosett treatment. However, this has a range of drawbacks, such as the release of adsorbable organic halogens (AOX) and the emission of chlorine compounds, which can harm the environment [3,4]. Therefore, the development of environmentally friendly wool shrink-proof technology has become a research focus and an area of difficulty in related fields.

Aiming to alter the directional friction effect (D.F.E.) and physical properties of wool fibers, researchers have developed a variety of shrink-proof finishing technologies, including biological enzyme treatment [5,6], plasma treatment [7,8,9], ultrasonic methods [10,11], and nanotechnological treatment [12]. However, due to problems related to insufficient finishing stability or high production cost, the above technologies still cannot be industrialized at present. Among them, the protease method is the most promising finishing technology to replace the traditional chlorination method, considering its environmental friendliness, high efficiency, and mild conditions [13,14,15,16,17,18]. There have been a number of studies on the application of various proteases to wool product finishing and keratin extraction. Li [19] used a keratinolytic protease (Proteinase K) to modify the wool fabric, improving its shrink-proof, anti-pilling, and dyeing properties. The results showed that Proteinase K preferentially cleaves the peptide and ester bonds in aromatic amino acids, sulfur-containing amino acids, and hydrophobic amino acids, which can hydrolyze the scale layer of wool effectively. Mei [20] covalently bound the protease molecule with poly (ethylene glycol) bis (carboxymethyl) ether (HOOC-PEG-COOH) and L-cysteine, in order to develop a novel tri-functional protease with reducibility, hydrolysis, and localization. Their experiment proved that the tri-functional protease can remove the scale layer on the wool surface under relatively mild conditions, efficiently achieving the effect of shrink-proof finishing. Zhang [21] used glutamine transaminase as a protein cross-linking enzyme, combined with air low-temperature plasma, in order to treat wool fibers. Kadam [22] coated the fabric with three polysaccharide biopolymers (arabic gum, wheat starch, and chitosan) through pad–dry–cure technology, following an enzyme. As compared to the untreated fabric (16.37%), the protease–chitosan combination showed the lowest area shrinkage (3.57%). After treatment, the wool fabric can satisfy the demand for machine washability. The above literature has proven that some biological enzymes can effectively peel the scale structure of wool, but the treatment process is typically complex and time-consuming.

There are many disulfide bonds in the scale layer of wool, which makes the wool surface less chemically reactive. Therefore, it is difficult to hydrolyze wool scales using enzymes alone, leading to the fact that protease shrink-proof finishing generally takes a long time [23,24,25]. To solve this problem, researchers often use protease in combination with other reagents or other techniques to improve the hydrolysis efficiency of protease on fibers. On one hand, the added auxiliary reagent should be able to significantly improve the hydrolysis efficiency of protease on wool, and on the other hand, it should not cause obvious harm to the environment. Therefore, selecting appropriate reagents and forming a complex system with protease is one of the key technologies to realize the industrialization of novel shrink-proof processes. In recent years, L-cysteine has attracted the attention of researchers in related fields. L-cysteine has a good reduction effect on disulfide bonds and is also non-toxic to organisms [26]; therefore, it has been widely considered in research, such as that related to the functional finishing of wool fibers and the extraction of keratin. Zhang [27] utilized L-cysteine as a reducing agent to dissolve wool fibers and prepare a keratin polypeptide solution. Their results showed that L-cysteine can destroy the disulfide bonds and effectively extract keratin. Du [28] pre-treated wool fabric with L-cysteine and then soaked it in a keratin polypeptide solution to achieve the purpose of surface modification. However, the treatment time in their study was too long, making the proposed process not suitable for production-scale application.

The various advantages of L-cysteine make it an ideal candidate for the shrink-proof finishing of wool. In this study, L-cysteine and protease 16L were formed into a treatment solution system, and the finishing of wool tops was carried out using the multiple-padding method. The finishing process flow is shown in Figure 1. We focused on determining the optimal process parameters of the L-cysteine/protease treatment solution system (L/PTSS) and the performance changes in the wool fiber after finishing. The effect of L-cysteine on the disulfide bonds in wool was explored by comparing the properties of L-cysteine and other four reducing agents: Tris (2-carboxyethyl) phosphine hydrochloride, sodium sulfite anhydrous, sodium bisulfite, and sodium hydrosulfite. Through the Response Surface Methodology, the optimal process parameters for L/PTSS were obtained. On this basis, the structure and properties of the treated fibers were characterized by scanning electron microscopy (SEM), ultraviolet spectral analysis, Fourier transform infrared analysis (FT-IR), and other testing technologies. The action mechanism and application value of L/PTSS are discussed in light of the results of the conducted experiments. The conclusions of this study are expected to provide a foundation for the development and industrial application of biological enzyme shrink-proof technology.

## 2. Results and Discussion

### 2.1. Selection of Reducing Agent in Compound Solution System

The ability of a reductant to treat wool fibers will directly affect the shrink-proof finishing effect of the protease solution system. In this paper, the properties of five reductants and their effects on wool were analyzed. Figure 2a–f shows the morphology of raw wool and fibers treated with different reductants. The intact scale layer can be seen on the surface of the untreated wool. After L-cysteine and Tris (2-carboxyethyl) phosphine hydrochloride (TCEP) treatment (Figure 2b,c), the scale structure was significantly damaged; in particular, the scales of TCEP-treated fibers were almost completely stripped. However, the other three reducing agents did not cause obvious hydrolytic damage to the scale structure of fibers under the same process conditions, and only incomplete etched dents could be observed on the scale surface. The SEM results illustrate that the various reducing agents led to different degrees of hydrolysis of the wool scale; thus, the selection of an appropriate agent will improve the shrink-proof finishing efficiency of wool.

Redox potential is used to reflect the redox properties of substances in an aqueous solution, where a negative potential indicates that the solution has certain reducibility. Through this method, the reduction performance of each reagent was indirectly compared. Figure 2g presents the redox potentials of the reducing agents under different pH values. In the results, the redox potentials of the five reducing agents were all negative and gradually decreased with the enhancement of pH value. At the same time, when the pH value was 8 (i.e., the pH value suitable for protease), the potentials of L-cysteine and TCEP were significantly lower than those of other reductants, indicating stronger reduction performance. This conclusion is consistent with the SEM results, demonstrating that L-cysteine and TCEP have a more significant reduction effect on wool under certain conditions.

The changes in the mechanical properties of wool fibers after reduction treatment are compared in Figure 2h. It can be seen that except for TCEP, the breaking strength of fibers treated with other reducing agents did not significantly decrease, and in some samples was even higher than in raw wool. Moreover, the elongation at break of wool after reduction treatment was also affected, which was generally reduced from 49.5% to about 30%. Among them, TCEP-treated fibers showed the most significant reduction in elongation at break, with an average value of only 24.1%. Comparing the results, it can also be seen that the dispersion of the strength results for the treated fibers was greater than that for the raw wool. This may be related to the uneven treatment effect of the reductants on wool fibers.

In summary, although both TCEP and L-cysteine can efficiently hydrolyze the scale structure of wool, as TCEP has a great impact on the mechanical properties of fibers, the solution system composed of L-cysteine and protease was selected for wool shrink-proof finishing in this study.

### 2.2. Statistical Modeling and Analysis

Table 1 lists the processing conditions used to optimize the shrink-proof finishing process and the performance characterization results of the treated fibers. In this model, the impacts of independent variables were assessed, including L-cysteine concentration, soaking time, and the padding number on response surfaces. The Design-Expert software was used to fit the response value of the breaking strength (Y_1_), the felt ball density (Y_2_), and the weight loss rate (Y_3_) through multiple regression. In addition, the strength of the raw wool was 6.15 cN and the density of the felt shrinkage ball was 135.86 kg/m^3^.

The model significance coefficients and variance analysis results for the breaking strength (Y_1_) are listed in Table 2. The regression model generated the relationship equation between breaking strength and independent variables, shown here as Equation (1). The confidence interval of Y_1_ at the 95% confidence level was (5.53, 6.26). In the equation, A, B, and C represent the values of L-cysteine concentration, soaking time, and padding number in the response surface methodology, respectively, and their value ranges are displayed in Section 3.4.
Y_1_ = 5.89 − 0.43 × A − 0.33 × B + 0.046 × C − 0.22 × AB + 0.032 × AC − 0.085 × BC − 0.25 × A^2^ + 0.60 × B^2^ − 0.12 × C^2^(1)

It can be seen from Table 2 that the *p*-value (0.0389) for the breaking strength regression model was less than 0.05, indicating that the model was significant (a *p*-value less than 0.05 indicates a significant difference, and a *p*-value less than 0.01 indicates an extremely significant difference) [29]. Among the linear terms, A and B were very significant, while item C was not significant. Combined with the F-value, it can be seen that among the three factors, the most influential factor on the fiber strength was the L-cysteine concentration (A), followed by the soaking time (B), and the least influential factor was the padding number (C). In addition, the model’s lack of fit *p*-value (0.036) was less than 0.05 and the R-value was only 0.839; these results indicate that the model did not fit the experimental results well. At the same time, interaction items (e.g., AB, AC, and BC) were also insignificant. These phenomena may be due to the strong discreteness of wool fibers. In follow-up research, it will be necessary to increase the number of fiber samples or optimize the calculation method for the experimental data, in order to improve the stability of the model.

The relevant results for the felt ball density (Y2) are presented in Table 3, and the relationship equation is shown as Equation (2). According to the software calculation, the 95% confidence interval was between 45.96 and 56.40. It can be seen from the table that the fitting model for felt ball density was a linear equation. As the *p*-value (0.0003) was less than 0.01, the model was extremely significant. In this model, L-cysteine concentration (A) had the most significant effect on felt ball density of wool, and soaking time (B) was also significant; however, the padding number (C) showed no significant effect. The results, with a lack of fit *p*-value greater than 0.05 and a smaller R^2^ value, indicated that the fitting degree of this model was also poor. This phenomenon may be caused by interference factors in the felt ball density experiment and fewer repetitions in each sample group.
Y_2_ = 53.18 − 12.21 × A − 4.93 × B − 4.28 × C (2)

Table 4 displays the ANOVA results for the weight loss rate. The equation generated by the model is shown as Equation (3). The 95% confidence interval for the weight loss rate was (10.07, 10.80). According to Table 4, the *p*-value was less than 0.01 in the regression model for weight loss rate, indicating that the model was extremely significant. For each influencing factor, B and C were extremely significant (*p* < 0.01), while A was significant (*p* < 0.05). Among them, the factor with the most obvious influence on the weight loss rate of fibers was the soaking time, and that with the lowest influence was the L-cysteine concentration. At the same time, the *p*-value for lack of fit was 0.1061 (*p* > 0.05) and the R^2^ value reached 0.963, demonstrating that Equation (3) was simulated well and could be used for further data analysis.
Y_3_ = 10.43 + 0.36 × A + 0.85 × B + 0.71 × C + 0.25 × AB + 0.56 × AC − 0.26 × BC + 1.42 ×A^2^ − 0.28 × B^2^ + 0.11 × C^2^
(3)

The effects of independent variable interactions on breaking strength and fiber weight loss rate are shown, in terms of response surfaces, in Figure 3. The highest center of the surface graph represents the extreme value for pairwise interactions [30]. The surfaces in Figure 3a,b,d,f, with large curvatures, indicate that the interactions between the process parameters were very significant. According to these surface models, the optimum process conditions were obtained through analysis in the Design-Expert software, as listed in Table 5. In conclusion, the process conditions of L/PTSS were determined as follows: the concentrations of L-cysteine and protease 16 L were 9 g/L and 1 g/L, respectively; the soaking time was 30 s; and the padding number was five times. In order to further verify the applicability of the model, we applied the optimal simulation process to finish wool fibers. The weight loss rate of fibers in the results was basically consistent with the predicted value, but there were certain deviations in the strength and the felt ball density. Although the mathematical model was found to be unstable for the prediction of results, it still has reference value for the optimization of process parameters, and the effectiveness of the developed model will be further improved in subsequent research.

### 2.3. Performance Analysis of Wool Fibers after Finishing

The structure and properties of the fibers treated with L/PTSS were tested, and the effect of this processing technology on wool was analyzed. The surface morphologies of raw wool fibers and the treated fibers are shown in Figure 4a,b. The raw wool presents a characteristic overlapping scale layers structure, where the edge of each cuticle shows a clear boundary. After treatment, the scales of the treated fibers were stripped, and the surface became relatively rough [31]. Figure 4c displays the felted ball shape of two fiber samples. On the left of the picture is the raw wool sample (135.86 kg/m^3^), while the fiber ball formed by the treated fibers (48.65 kg/m^3^) is shown on the right. It can be seen that under the same conditions, the volume of felted balls formed by the treated fibers was larger, while the density was relatively low. Iglesias et al. [23] used a biosurfactant extracted from *Bacillus subtilis* for wool pre-treatment, and then utilized an extracellular proteolytic extract of Bacillus to perform shrink-proof finishing on wool fibers. The felt ball density results indicated that this method can significantly reduce the fiber felting tendency without a significant loss in wool tensile strength. The felting ball density of the treated wool in their study was 49 kg/m^3^, similar to the density value reported in the present study. The results indicate that the felting shrinkage of fibers after finishing was effectively improved.

The thermal stability of fibers can be determined by thermogravimetric analysis (Figure 4d,e). According to the test results, the thermal weight loss of raw wool was mainly divided into two stages. The first stage occurred in the range of 0–100 °C, with the weight loss ratio reaching about 6%, mainly caused by the evaporation of water from fibers. The second stage was the thermal degradation of wool at 220–450 °C [32,33], in which 62.7% of the thermal weight loss occurred, where the degradation rate was the fastest at 337.17 °C. When the test temperature reached 600 °C, the mass residue ratio of raw wool was only 21.75%. The thermal degradation law of treated fibers was similar to that of raw wool, but the mass residue ratio (21.75%) and the fastest degradation temperature (318.65 °C) were reduced under high-temperature conditions. These results illustrate that the thermal stability of wool was decreased, to a certain extent, after the scale structure was hydrolyzed by the L/PTSS, but the fibers still possessed adequate thermal properties.

Figure 4f compares the mechanical property changes in raw wool and treated fibers. Clearly, after finishing, the tensile breaking strength of wool decreased from 6.15 cN to 5.88 cN, and the strength retention was as high as 95.6%. In contrast, the elongation at break decreased significantly, which reduced from 49.5% when untreated to about 28.8%. Combined with the above conclusions, it can be determined that the L/PTSS effectively destroyed the scale structure to achieve the effect of shrink-proof finishing. At the same time, the thermal and mechanical properties of wool were not excessively affected, thus satisfying certain production and application requirements.

### 2.4. Shrink-Proof Finishing Mechanism of Wool Fibers by L/PTSS

To explore the shrink-proof law and mechanism by which L/PTSS acts on wool, a variety of techniques were employed to analyze the chemical composition and structural changes of the fiber samples. XPS was carried out to determine the elemental content and valence change on the surface of wool in the range of 3–5 nm. Figure 5a shows the content results of four main chemical elements in wool. The C content of the raw fiber was 80.71%, which was more than that in whole wool (50–55%), due to the presence of a lipid layer on the fiber surface. The C content on the fiber surface decreased to 78.86% after the treatment, while the content of N on treated fibers (5.89%) was higher than that on raw wool (3.35%), and the proportion of S on the surface decreased from 1.55 when untreated to 1.14 [34,35]. The main reason for the elemental content change is that parts of the scale structure are destroyed by hydrolysis during the finishing process, exposing polar groups such as amino groups on the fiber surface. From Figure 5b,c, we can analyze the valence state of sulfur on the fibers after finishing through the binding energy results. The spectrum can be fitted to two peaks, 163.69 eV and 166.20 eV, corresponding to -S–S- and -S–O- bonds, respectively [36]. Compared with raw wool, the intensity value of the -S–S- characteristic peak in treated fibers decreased from 2131 to 1993, while the -S–O- characteristic peak increased noticeably. These phenomena prove that the L/PTSS reduced the disulfide bonds in wool scale to form thiol groups during the shrink-proof finishing process, where some thiols were further oxidized by the environment to form sulfonate, resulting in decreased disulfide bond content in the treated fibers [35,37].

The content change of specific amino acids in the finishing residual liquid was measured through a UV absorbance test in order to indirectly analyze the hydrolysis effect of the L/PTSS on the scale structure. When the chemical compound contains conjugated double bonds, it will absorb ultraviolet rays at certain wavelengths. Therefore, some amino acids containing benzene rings will form characteristic peaks in the UV absorption spectrum. Among them, phenylalanine, tyrosine, and tryptophan will form absorption peaks at 258, 275, and 280 nm, respectively [38]. Figure 5d presents the UV absorption spectra of the residual liquid after each padding. It can be seen that the residual liquid remaining in different padding tanks formed absorption peaks in the regions of 250–260 and 270–290 nm, consistent with data in the literature. Among them, the absorption peak at 250 nm was relatively weak, while the peak at 280 nm was strong, which should be formed by the superposition of the characteristic peaks of tyrosine and tryptophan. Meanwhile, the absorption peak intensity of the residual liquid in each tank increased sequentially, where the intensity for the second tank was enhanced the most significantly compared with the first one. The results of the UV absorbance test illustrate that the L/PTSS had a significant hydrolysis effect on wool, and the degree of hydrolysis was improved with an increase in the padding number. In addition, the scale structure of wool in the first tank was relatively dense, and L-cysteine was required to react with disulfide bonds to soften the scales. At this time, the hydrolysis of protease on fibers is still weak. When the fibers enter the second tank, the protease can hydrolyze the scale layer more efficiently. Therefore, the content of the hydrolyzed peptide chains in the second tank changed significantly.

Raman and infrared spectroscopy can be utilized to explore the changes in the chemical and aggregation structures of wool after finishing. Figure 5e shows the Raman spectra of different samples, from which it can be seen that the peak positions in raw wool and treated fibers were similar. The characteristic peaks of disulfide bonds were mainly distributed at 511 cm^−1^. In order to estimate the effect of the shrink-proof process on the disulfide bonds, we calculated the strength ratio for the -S–S- peak and the peak distributed at 1451 cm^−1^, according to Equation (4) [39]. The larger the value of R, the higher the content of disulfide bonds on wool surface. After calculation, it was found that the R-value of the treated fibers was 0.337, significantly smaller than that of the raw wool (0.452). This result proves again that the solution system had a significant chemical action on the disulfide bonds in wool. In addition, a new characteristic peak appeared at 669 cm^−1^ in the spectrum of the treated fibers. According to the literature, this peak was mainly caused by the stretching vibration of the C–S bond [40]. The increase in C–S bond content may be related to the destruction of disulfide bonds and the formation of sulfonate radicals. This may also have been caused by the bonding of L-cysteine to some thiol or hydroxyl groups in wool.
R = I_511_/I_1451_ (I stands for the peak intensity value)(4)

The Fourier infrared spectrum is presented in Figure 5f, in which the main peaks representing the characteristics structure of peptide bonds are located at 3277 cm^−1^, 1631 cm^−1^, 1514 cm^−1^, and 1237 cm^−1^. The broad band from 3300 cm^−1^ to 3200 cm^−1^ represents amide A, caused by stretching vibrations of the N–H bonds in peptide chains. The peak found at 1680–1610 cm^−1^ is associated with the stretching vibration of C=O (amide I) [41]. In addition, the amide II and amide III absorption peaks of wool were both located near 1523 cm^−1^ and 1232 cm^−1^, representing the bending vibration of N–H bonds and the stretching vibration of C–N bonds [42,43]. Compared with raw wool, the spectrum of the treated fibers was not changed significantly, and only the characteristic peak of amide I had a certain redshift. This change could have been caused by a small change in the proportion of fiber crystal structure after the scales were partially stripped [44].

The chemical structure transformation of wool can be further analyzed by measuring the hydrolyzed amino acid content. The composition results for raw wool and treated fibers are shown in Figure 5g, in terms of the mass percent of each kind of amino acid. Among them, cystine had the most significant change in content, which was reduced from 10.09% in raw wool to 3.80% after treatment. This result is consistent with the XPS analysis, showing that the L/PTSS effectively cleaved the disulfide bond in cystine and further hydrolyzed the scale structure [45]. In addition, only alanine, glutamic acid, and aspartic acid increased among the other amino acids. This phenomenon may have been caused by the different proportions of amino acids between the scale layer and the cortex layer, or the reduction in cystine content.

In summary, based on effectively reducing the disulfide bonds in wool, L/PTSS achieved the effect of rapidly destroying the scale layer through the hydrolysis reaction of protease and the physical action of multiple padding, thereby realizing the shrink-proof finishing of wool fibers.

## 3. Materials and Methods

### 3.1. Materials

Wool fibers (70s Australian wool, diameter 18–20 μm) were provided by Zhejiang Xinao Textile Inc. (Tongxiang, China). L-cysteine, Tris(2-carboxyethyl) phosphine hydrochloride (TCEP), and Quinhydrone were purchased from Shanghai Aladdin Biochemical Technology Co., Ltd. (Shanghai, China). Sodium hydroxide (NaOH) was obtained from Tianjin Tianli Chemical Reagent Co., Ltd. (Tianjin, China). Protease 16L was supplied by Novozyme Biotechnology Co., Ltd. (Tianjin, China). Sodium sulfite anhydrous (Na_2_SO_3_), sodium hydrosulfite (Na_2_S_2_O_4_), sodium phosphate monobasic dihydrate, and disodium hydrogen phosphate dodecahydrate were acquired from Tianjin Damao Chemical Reagent Factory (Tianjin, China). Sodium bisulfite (NaHSO_3_) was obtained from Tianjin Beilian Fine Chemicals Development Co., Ltd. (Tianjin, China). Potassium hydrogen phthalate was received from Shanghai INESA Scientific Instrument Co., Ltd. (Shanghai, China).

### 3.2. Effect Analysis of Different Reducing Agents on Wool Fibers

Five reducing agents (L-cysteine, Na_2_SO_3_, NaHSO_3_, Na_2_S_2_O_4_, and TCEP) were selected to treat wool fibers, and the effects of each agent on the structure and properties of wool were studied. The fibers were soaked in the treatment solution with a concentration of 0.5 M at 50 °C for 1 h. The bath ratio was set to 1:50, and the pH value of each finishing liquid was adjusted to 8 using phosphate buffer solution. After the treatment, the fibers were washed and dried at 80 °C for 12 h before testing.

### 3.3. Shrink-Proof Finishing of Wool with L/PTSS

The treatment solution system was composed of a reducing agent and 16 L protease, and the shrink-proof finishing process of wool fibers is shown in Figure 1. Washed wool tops were wetted with 1.5 g/L penetrant solution (fatty alcohol–polyoxyethylene ether) at 50 °C for 30 s. Then, the wool samples were soaked for a certain time with L/PTSS at 50 °C at a pH of 8.0, followed by squeezing with a padder (Laizhou Yuanmao Instrument Co., Ltd., Shandong, China). The treatment operation was repeated several times, where the soaking time was 30–60 s each time. After finishing, the wool tops were rinsed with water at 50 °C for 60 s, and the enzyme was inactivated with water at 80 °C for 30 s.

### 3.4. Response Surface Methodology (RSM) Analysis

The Design-Expert.V8.0.6.1 software was used for RSM design, according to the Box–Behnken design (BBD). The main factors of the pre-treatments and shrink-proof finish procedures were chosen through preliminary investigations. The details of the treatment design were generated using 3 × 3 factor coding, as presented in Table 6.

### 3.5. Measurements

#### 3.5.1. Tensile Breaking Strength of Fibers

The fibers treated by different processes and the raw wool were fully washed with distilled water, then dried in an oven at 80 °C for 24 h. The fiber samples were stored in a standard atmospheric environment (temperature, 20 ± 2 °C; relative humidity, 65 ± 2%) for 2 days before testing [46]. The breaking strength and elongation at break were measured using a YG003A single-fiber strength machine (Taicang Textile Instrument Factory, Taicang, China). A total of 50 single fibers were tested with a pre-tension of 0.1 cN, gauge length of 10 mm, and clamp speed of 10 mm/min. We calculated the average strength and elongation of each group, and took these values as the final results for the fiber samples.

#### 3.5.2. Scanning Electron Microscopy (SEM)

The fiber samples, after drying, were sputter-coated with gold under a vacuum with a current of 10 mA, then observed using a QUANTA-450-FEG Field Emission Scanning electron microscope (FEI Corporation, Hillsboro, OR, USA) at 5000× magnification and with an accelerating voltage of 10 kV.

#### 3.5.3. Redox Potential Test

An ORP electrode (INESA Scientific Instrument Co., Ltd., Shanghai, China) was used to test the redox potential of the reductant solution under different conditions. We dissolved a certain amount of potassium hydrogen phthalate in 250 mL water to prepare a buffer solution with pH value of 4. The ORP standard solution was prepared by adding 0.5 g hydroquinone to 50 mL buffer solution and dissolved at 25 °C. The ORP electrode was immersed in the solution until the data were stable, in order to obtain the redox potential of the analyte liquor.

#### 3.5.4. Fiber Weight Loss Ratio

The fibers were dried at 80 °C for 12 h and weighed until a constant weight was reached. The weight of samples was recorded as M_0_ before the anti-felting treatment and M after treatment and drying. Wool samples were weighed using an FA2004B electronic balance (Shanghai Anting Scientific Instrument Factory, Shanghai, China), and the weight loss ratio was calculated according to Equation (5):(5)$$\delta =\frac{M_{0}-M}{M_{0}}\times 100\%,$$
where δ denotes the weight loss ratio (%), M is the wool weight (g) of each group after treatment and drying, and M_0_ is the mass (g) of the fibers without treatment.

#### 3.5.5. Felt Ball Density Analysis

Wool was dried at 80 °C and stored in a standard atmosphere environment for 2 days before and after testing. The fibers were weighed to 1.0 g and teased loosely and straight. Then, the samples were immersed in dye pots with 50 mL water and stirred to wet them completely. We placed the dye pots into an Infrared dyeing machine (Xiamen Rapid Precision Machinery Co., Ltd., Xiamen, China) and treated them at 40 °C for 30 min with a heating rate of 1 °C/min.

After the test, the felt ball diameters were measured using vernier calipers; readings were made to the nearest 0.1 cm. The volume and density of the felt balls were calculated according to Equations (6) and (7), respectively:(6)$$V=\frac{\pi }{6}\times a\times b\times c,$$
(7)$$\rho =\frac{m}{V}\times 1000,$$
where V denotes the volume of the felt ball (cm^3^); a, b, and c are the long axis, short axis, and height (cm) of the felt shrinkage ball, respectively; m is the weight (g) of the wool fibers; and ρ denotes the felt ball’s density (kg/m^3^).

#### 3.5.6. Ultraviolet (UV) Spectrum

The residual liquid in different finishing tanks was filtered and stored in centrifuge tubes. Using the mixed shrink-proof finishing liquid as the blank control group, the UV spectra were recorded using a T3202 ultraviolet–visible spectrophotometer (Shanghai Youke Instrument Co., Ltd., Shanghai, China) in the wavenumber range of 250–320 nm at a scanning frequency of 50 Hz.

#### 3.5.7. Fourier Transform Infrared Spectroscopy (FT-IR)

Fourier infrared scanning spectra of raw wool and treated fibers were investigated using a Nexus670 Fourier infrared spectrometer (Thermo Fisher Scientific, Waltham, MA, USA). The spectra were obtained in the wavenumber range of 600–4000 cm^−1^ at a resolution of 4 cm^−1^. All spectra were baseline corrected.

#### 3.5.8. Thermogravimetric Analysis (TG)

The thermal degradation performance of wool fibers was measured using a TGA2 thermogravimetric analyzer (Mettler Instrument Company, Greifensee, Switzerland). Fiber samples were tested using a ceramic crucible from 50 to 600 °C at a rate of 10 °C/min under a nitrogen atmosphere.

#### 3.5.9. X-ray Photoelectron Spectroscopy (XPS)

XPS spectra were recorded using a Thermo Scientific K-Alpha photoelectron spectrometer (Thermo Fisher Scientific, USA) fitted with an Al K Alpha radiation (1486.6 eV) under the following conditions: spot size, 400 µm; pass energy, 50 eV; and step size, 0.1 eV. The high voltage was kept at 12.0 kV.

#### 3.5.10. Raman Spectra

A LabRAM HR Evolution Raman spectrometer (Horiba, France) was utilized to test the conformation of protein molecules in the wool fibers. The laser beam on the sample was focused to a spot diameter of 1 µm under a 100× microscope objective. Spectra were recorded by scanning the 400–2000 cm^−1^ region with a selected laser having wavelength of 514 cm^−1^.

#### 3.5.11. Amino Acid Content Analysis

The amino acid contents of fibers were determined using a Biochrom30 + amino acid analyzer (Biochrom, Cambridge, UK). The samples were hydrolyzed in 6 M hydrochloric acid (HCL) for 24 h at 110 °C under a nitrogen atmosphere. Hydrolyzed amino acid residues were derived from hydroxyl succinimidyl carbamate (AQC, Waters, Milford, MA, USA) and eluted on a reversed-phase column. An Alliance High-Performance Liquid Chromatograph (HPLC) (Waters, USA) was used, and the eluate was detected at 0.22 μm. The quantitative amino acid composition (expressed as mol% for each amino acid) was determined by external standard calibration (Amino Acid Standard H, Pierce).

## 4. Conclusions

In the present work, we developed an environmentally friendly wool shrink-proof finishing technology, in which we considered L-cysteine and protease 16 L to form a treatment solution system, and where the wool fibers were treated using the padding finishing method. The reducing abilities of L-cysteine and four other reducing agents were compared through a redox potential test. The wool was treated with these reagents at 50 °C for 1 h, and the reaction effect of the fiber was analyzed. The results demonstrated that under a certain pH value, L-cysteine displayed stronger reduction performance and a significant destructive effect on wool scales, indicating its suitability for research on shrink-proof finishing of wool. The response surface method was used to determine the optimal finishing parameters of L/PTSS. The obtained process conditions were as follows: the concentrations of L-cysteine and protease 16 L were 9 g/L and 1 g/L, respectively; the wool fibers were padded five times at 50 °C; and the immersion time was 30 s. On the basis of determining these process conditions, SEM, felt ball density analysis, TG, and tensile breaking strength tests were carried out to study the structure and property changes of wool fibers after shrink-proof finishing. The comprehensive analysis demonstrated that the scale layer of wool was stripped evenly, while the fibers still retained adequate thermal and mechanical properties. Meanwhile, in order to deeply study the action mechanism of L/PTSS on wool, the chemical properties of fibers were characterized by XPS, FT-IR, Raman, and other techniques. The conclusions proved that L-cysteine can rapidly reduce the disulfide bonds in wool, thus improving the reactivity of the scale structure. After that, the protease can more easily contact the reaction site in the scale to hydrolyze the keratin polypeptide chain. Compared with other protease shrink-proof methods, the L/PTSS technology developed in this study addresses the problem of the slow action rate of protease on wool by reasonably designing the finishing solution system and processing method. After treatment, the shrink-proof property of the wool fiber was significantly improved, while its mechanical properties were not excessively damaged. At the same time, the reagents used in this method are environmentally friendly materials, indicating broad development prospects and research value. Follow-up research will focus on further optimizing the L/PTSS finishing process, in order to ensure that wool of different qualities can reach the machine-washable industry standard after treatment, gradually promoting the industrial application of this technology.

## Figures and Tables

**Figure 1 ijms-23-13553-f001:**
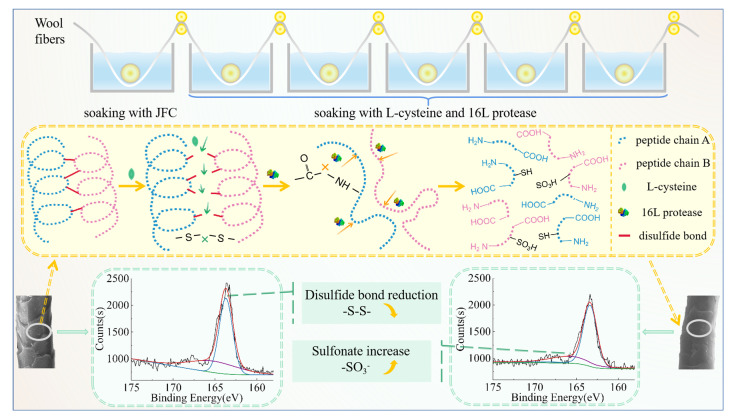
Schematic diagram showing the technological process and reaction mechanism of shrink-proof finishing with L/PTSS.

**Figure 2 ijms-23-13553-f002:**
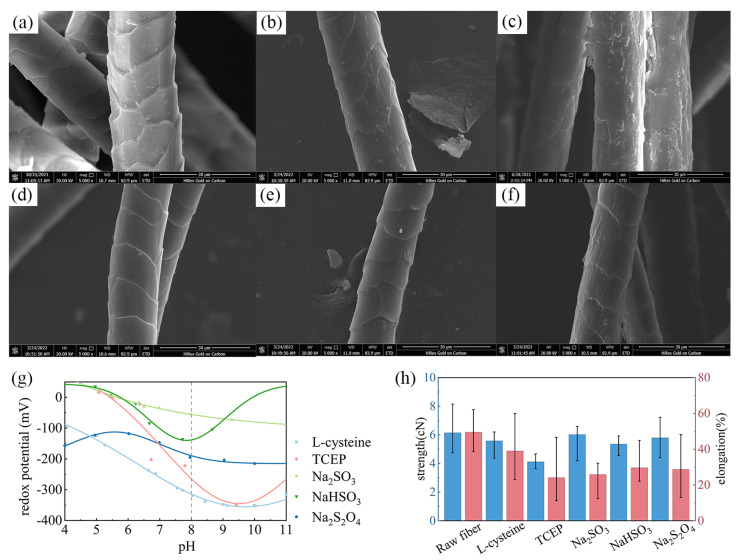
SEM images of wool fibers treated with different reductants: (**a**) raw fibers; (**b**) L-cysteine; (**c**) TCEP; (**d**) Na_2_SO_3_; (**e**) NaHSO_3_; (**f**) Na_2_S_2_O_4_. (**g**) The redox potential of different reductant solutions; (**h**) the breaking strength and elongation of fiber samples.

**Figure 3 ijms-23-13553-f003:**
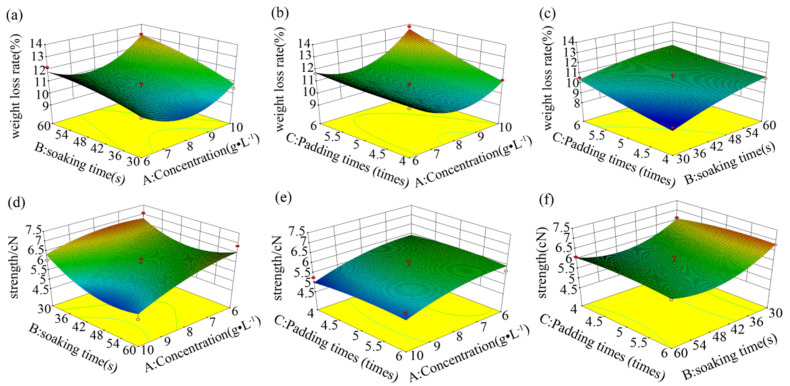
Response surfaces of the factor interaction relationships for the weight loss rate (**a**–**c**) and breaking strength (**d**–**f**).

**Figure 4 ijms-23-13553-f004:**
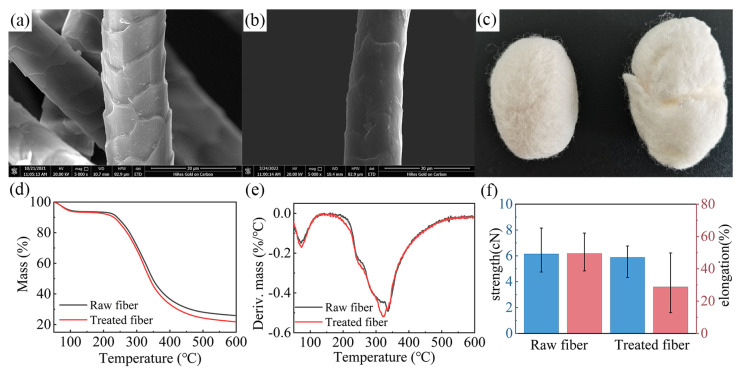
SEM images of (**a**) raw wool and (**b**) treated fibers. (**c**) Felted ball method test results. (**d**) TG, (**e**) DTG, and (**f**) tensile strength results of the fiber samples.

**Figure 5 ijms-23-13553-f005:**
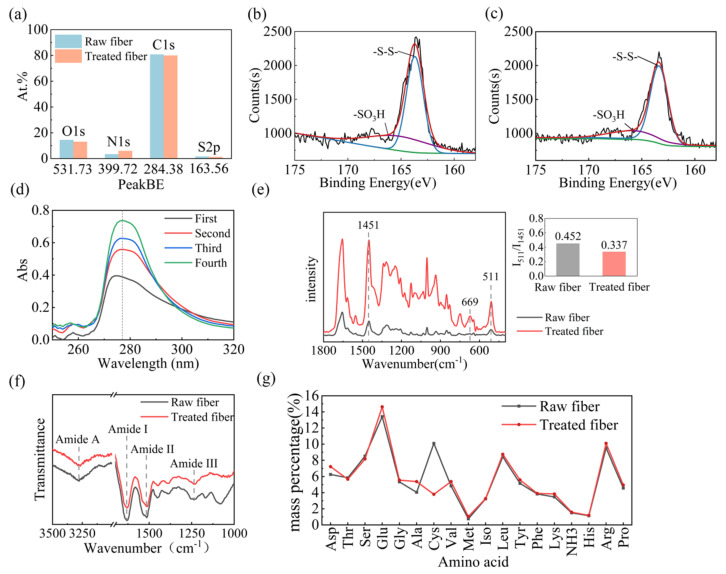
(**a**) The content changes of main elements on the surface of different fiber samples. Binding energy spectra of sulfur in raw wool (**b**) and treated fibers (**c**). (**d**) UV absorption spectra of the residual solution in each soaking tank after finishing wool fibers with L/PTSS. (**e**) Raman spectra, (**f**) Fourier infrared spectra, and (**g**) amino acid content of raw wool and treated fibers.

**Table 1 ijms-23-13553-t001:** Experimental design and responses of the shrink-proof finishing process.

Sample	Concentration of L-Cysteine (g·L^−1^)	Soaking Time (s)	Padding Number (Times)	Breaking Strength (cN)	Felt Ball Density (kg·m^3^)	Weight Loss Rate (%)
1	6	30	5	7.07	78.5	10.52
2	6	45	6	5.71	60.63	11.68
3	8	30	6	6.8	56.91	10.56
4	10	45	6	5.5	35.77	13.79
5	8	60	4	6.11	58.83	10.49
6	6	45	4	5.6	64.16	11.25
7	8	45	5	5.84	53.42	10.3
8	6	60	5	6.86	54.98	12.19
9	10	60	5	4.97	45.17	13.14
10	8	45	5	5.67	44.11	10.24
11	8	45	5	5.91	50.97	10.37
12	10	45	4	5.26	40.6	11.13
13	8	45	5	6.15	49.67	10.82
14	10	30	5	6.04	39.05	10.45
15	8	60	6	5.95	45.53	11.27
16	8	30	4	6.62	69.5	8.73
17	8	45	5	5.89	56.22	10.43

**Table 2 ijms-23-13553-t002:** ANOVA results for breaking strength.

Variance Source	Sum of Squares	df	Mean Square	F-Value	*p*-Value
Model	4.370	9	0.490	4.0700	0.0389
A (Concentration)	1.510	1	1.510	12.610	0.0093
B (Soaking time)	0.870	1	0.870	7.300	0.0306
C (Padding number)	0.017	1	0.017	0.140	0.7162
AB	0.180	1	0.180	1.550	0.2533
AC	0.004	1	0.004	0.035	0.8561
BC	0.029	1	0.029	0.240	0.6377
A^2^	0.270	1	0.270	2.290	0.1740
B^2^	1.500	1	1.500	12.610	0.0093
C^2^	0.060	1	0.060	0.510	0.4999
Residual	0.840	7	0.120		
Lack of fit	0.720	3	0.240	8.040	0.0361
Pure error	0.120	4	0.030		
Cor total	5.200	16			
R^2^	0.839		Adj R^2^	0.633	

**Table 3 ijms-23-13553-t003:** ANOVA results for felt ball density.

Variance Source	Sum of Squares	df	Mean Square	F-Value	*p*-Value
Model	1533.84	3	511.28	13.53	0.0003
A (Concentration)	1192.67	1	1192.67	31.56	<0.0001
B (Soaking time)	194.54	1	194.54	5.15	0.0409
C (Padding number)	146.63	1	146.63	3.88	0.0705
Residual	491.21	13	37.79		
Lack of fit	408.94	9	45.44	2.21	0.2314
Pure error	82.27	4	20.57		
Cor total	2025.06	16			
R^2^	0.757		Adj R^2^	0.702	

**Table 4 ijms-23-13553-t004:** ANOVA results for weight loss rate.

Variance Source	Sum of Squares	df	Mean Square	F-Value	*p*-Value
Model	21.51	9	2.390	19.96	0.0003
A (Concentration)	1.03	1	1.030	8.60	0.0219
B (Soaking time)	5.83	1	5.830	48.71	0.0002
C (Padding number)	4.06	1	4.060	33.92	0.0006
AB	0.26	1	0.260	2.17	0.1840
AC	1.24	1	1.240	10.38	0.0146
BC	0.28	1	0.280	2.30	0.1730
A^2^	8.51	1	8.510	71.07	<0.0001
B^2^	0.33	1	0.330	2.73	0.1426
C^2^	0.05	1	0.050	0.42	0.5386
Residual	0.84	7	0.120		
Lack of fit	0.63	3	0.210	4.02	0.1061
Pure error	0.21	4	0.052		
Cor total	22.35	16			
R^2^	0.963		Adj R^2^	0.914	

**Table 5 ijms-23-13553-t005:** Model predicted parameters for optimal finishing process and the results of the validation experiment.

	Concentration of L-Cysteine (g·L^−1^)	Soaking Time (s)	Padding Number (Times)	Breaking Strength (cN)	Felt Ball Density (kg·m^3^)	Weight Loss Rate (%)
Prediction	9.00	30	5.34	6.68	50.55	10.15
Verification	9.00	30	5	5.88	48.65	10.45

**Table 6 ijms-23-13553-t006:** Process variables and experimental levels.

Factor	Coding	Level		
		−1	0	1
Concentration of L-cysteine (g·L^−1^)	A	6	8	10
Soaking time (s)	B	30	45	60
Padding number (times)	C	4	5	6

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
