# Peer review of "Development of Environmentally Friendly Wool Shrink-Proof Finishing Technology Based on L-Cysteine/Protease Treatment Solution System"

_ijms, 2022, doi:10.3390/ijms232113553_

Round 1
Reviewer 1 Report
The research work presented in the manuscript was involved in the investigation of wool treatment with four different reducing agents. It is interesting to understand the redox potential of these different reducing agents and their effect on the wool fibre properties. This knowledge was leading to the research of wool shrink-resist finishing using the combination of L-Cysteine and enzyme protease. Different mechanical and analytical testing methods were used for monitoring the wool fibre properties during the treatment including tensile strength and felting shrinkage. However, there are some issues raised. Clear results are needed to confirm how wool shrink-resistance can be achieved by the enzyme process with protease and L-cysteine without causing significant fibre damage.
Herewith the major comments for authors to consider and improve the manuscript.
1) Wool fibres don’t have identical fibre diameter in the cross-sections. The fibre diameters of wool sample are shown in the distribution over a range of diameters. The fibre strength should be directly related to their fibre diameters. The research work showed the average tensile strength of wool sample from the measurement of 50 individual fibres. The results in the figures should include their deviation for the tensile strength of 50 individual fibres tested. Without this information, it is difficult to tell the significance in the effect of treatment with L-Cysteine on the fibre strength.
2) The amino acid content analysis was used. The results were discussed to show the reduction of cystine content for the wool samples treated with protease and L-cysteine. Authors explained the cleavage of disulphide bonds in wool caused the reduction of cystine content. How about the cysteine residues?
3) In Figure 1, schematic diagram shows the enzymatic process through repeat dippings and paddings. It is not clear about the soaking time and padding times shown in Table 1. The soaking time is for each dipping. With the number of padding increased, the soaking time is also multiple of each soaking time. Is this correct?
4) Authors should provide full form for the first use of the abbreviation. For example: TCEP at line 108 of page 3. Without reading further, it is not understandable for A, B, and C used in the Equation 4
5) SEM images are not clear.
6) Authors used “composite” to describe the mixture of L-Cysteine and protease used in the treatment solution. The word of “composite” is not appropriate.
7) “dye vat”?
8) Page 11/L322, …..Na2S2O4 and Na2S2O4… same?
9) Page 11/Line 331: what is “penetrant solution”?
10) Based on the model prediction, has the optimum finishing process been applied on the wool treatment to confirm the predicted results?
Author Response
Response to Reviewer 1 Comments
Dear editors and reviewers:
First of all, thank you for your kindly and hard work for our paper. We are very grateful to have been given the opportunity to revise our manuscript. We have carefully considered the comments and modified the manuscript accordingly. Revised portion were marked up using the “Track Changes” function and we hope the improved manuscript will meet your magazine’s standard. The main corrections and the responds are listed as follow:
Point 1: Wool fibres don’t have identical fibre diameter in the cross-sections. The fibre diameters of wool sample are shown in the distribution over a range of diameters. The fibre strength should be directly related to their fibre diameters. The research work showed the average tensile strength of wool sample from the measurement of 50 individual fibres. The results in the figures should include their deviation for the tensile strength of 50 individual fibres tested. Without this information, it is difficult to tell the significance in the effect of treatment with L-Cysteine on the fibre strength.
Response 1: Thank you for your constructive comment. Based on your suggestion, we have reorganized the strength data for each fiber samples and reflected the degree of deviation in the test results with error bars. At the same time, we have also made corresponding modifications to the analysis of fiber mechanical properties in Figures 2 and 4.
Point 2: The amino acid content analysis was used. The results were discussed to show the reduction of cystine content for the wool samples treated with protease and L-cysteine. Authors explained the cleavage of disulphide bonds in wool caused the reduction of cystine content. How about the cysteine residues?
Response 2: It is thankful for the recommendation. The amino acid analysis equipment used in this study mainly uses ninhydrin colorimetry to determine the content of each amino acid. Each amino acid in this test will be measured with an external standard calibration, and only the content data of some specific amino acids can be obtained. Among them, the content of cysteine cannot be achieved by this testing.
Point 3: In Figure 1, schematic diagram shows the enzymatic process through repeat dippings and paddings. It is not clear about the soaking time and padding times shown in Table 1. The soaking time is for each dipping. With the number of padding increased, the soaking time is also multiple of each soaking time. Is this correct?
Response 3: Thank you for your careful comment. We are very sorry for the unclear expression of some content in the article. The padding times in this paper refers to the number of shrink-resistant finishing tanks. For example, 5 times of padding means that the wool top passes through five finishing tanks filled with L-cysteine and protease mixture in turn. Soaking time, as you say, refers to the duration of the wool is dipped in each finishing tank. The change of padding times has no effect on the soaking time.
Point 4: Authors should provide full form for the first use of the abbreviation. For example: TCEP at line 108 of page 3. Without reading further, it is not understandable for A, B, and C used in the Equation 4.
Response 4: Thank you for your valuable advice. According to your suggestion, we have given the full name of the abbreviations and the meaning of letters that appear for the first time in the paper, including TCEP and the letters in formula 4.
Point 5: SEM images are not clear.
Response 5: Thank you for your careful comment. We are so sorry for these mistakes in Figures. According to your suggestion, we re-processed the electron micrographs in Figure 2 and 4 to give them better resolution. Then these modified photos are used in the revised manuscript.
Point 6: Authors used “composite” to describe the mixture of L-Cysteine and protease used in the treatment solution. The word of “composite” is not appropriate.
Response 6: Thank you for your valuable advice. After reviewing the relevant literatures, we changed the "composite solution" in the article to "combined solution", to ensure that the wording in manuscript is more reasonable and correct.
Point 7: “dye vat”?
Response 7: We are very sorry for some inappropriate expression in articles. In the felt ball density analysis, the infrared sample dyeing machine we used is similar to the equipment structure in the following figure. During the experiment, the fiber samples and the finishing solution were stored in the dye cup for constant temperature finishing. With your reminder, we realized that it is not appropriate to use "dye vat" here, so we revised it to "dye cup" in the manuscript.
Point 8: Page 11/L322,…..Na2S2O4 and Na2S2O4… same?
Response 8: Thank you for your careful comment. We have revised the error you pointed out in our revised manuscript. The revisions are committed as followed:
Five reducing agents (L-cysteine, Na2SO3, NaHSO3, Na2S2O4, and TCEP) were selected to treat wool fiber, and the effects of each agent on the structure and properties of wool were studied.
Point 9: Page 11/Line 331: what is “penetrant solution”?
Response 9: We are very sorry for some illegibility in the article. The "penetrant" in manuscript refers to fatty alcohol–polyoxyethylene ether solution, which is used to ameliorate the wettability of wool fiber surface and improve the efficiency of shrink proof finishing. We have added the specific compound name to the paper, and the revised section as followed:
Washed wool tops were wetted with 1.5g/L penetrant solution (fatty alcohol–polyoxyethylene ether) at 50℃ for 30s.
Point 10: Based on the model prediction, has the optimum finishing process been applied on the wool treatment to confirm the predicted results?
Response 10: Thank you for your valuable advice. In the original manuscript, after obtaining the optimal process through the RMS mathod, the properties of the fibers treated under this condition were characterized. These include the tensile strength, the felt ball density and the weight loss rate results of the samples. However, some test results did not directly compare with the predicted results. According to your suggestion, we have added the experimental values treated with the optimal process in Table 5 and compared them with the predicted values. The results show that although the predicted value of the mathematical model has a certain deviation from the experimental results, the model still has reference value for the optimization of the shrink-proof finishing process.
Special thanks to you for your helpful comments.

Reviewer 2 Report
This work reports wool shrink proofing using L-Cysteine – Protease combination. The manuscript is well written and presents all required parameters in detail. The manuscript may be accepted after a revision as suggested below:
1. Line 42 – 45: The sentence “At present, the most commonly used finishing method ….. damage to the environment and humans [3,4]” needs rewriting. The suggested replacement statement is: “Traditionally, the most commonly used finishing in industrial production is Chlorine-Hercosett treatment. However, it has a range of drawbacks like releases of adsorbable organic halogens (AOX) and emission of chlorine compounds that harm the environment”.
2. The literature review is not comprehensive. Authors should report other key methods recently studied.
Suggestion: https://doi.org/10.1016/j.susmat.2021.e00298
3. Authors should highlight the advantage of L-Cysteine – Protease combination over other methods.
4. Line 57 protease k, “K” should be capitalized.
5. Line 93 Kindly mention the name of the reducing agents
6. Line 99-100: Repeating the last statement of the abstract
7. The material and methods section should be before the results section. Please check with journal style.
8. Line 345-346: Check English and grammar: Authors should say what and how they have processed their samples.
9. Line 355: “electron microscopy” should be “Electron Microscopy”
10. Line 443: Replace “shrunk” with “shrink”
11. Equation numbers throughout the manuscript are not consistent and not in line.
12. Please cross-check the values of the felt ball test compared with the existing literature.
Author Response
Response to Reviewer 2 Comments
Dear editors and reviewers:
First of all, thank you for your kindly and hard work for our paper. We are very grateful to have been given the opportunity to revise our manuscript. We have carefully considered the comments and modified the manuscript accordingly. Revised portion were marked up using the “Track Changes” function and we hope the improved manuscript will meet your magazine’s standard. The main corrections and the responds are listed as follow:
Point 1: Line 42 – 45: The sentence “At present, the most commonly used finishing method ….. damage to the environment and humans [3,4]” needs rewriting. The suggested replacement statement is: “Traditionally, the most commonly used finishing in industrial production is Chlorine-Hercosett treatment. However, it has a range of drawbacks like releases of adsorbable organic halogens (AOX) and emission of chlorine compounds that harm the environment”.
Response 1: Thank you for your valuable advice. According to your suggestion,the sentence “At present, the most commonly used finishing method ….. damage to the environment and humans [3,4]” has been rewritten in the introduction section of revised manuscript.
Point 2: The literature review is not comprehensive. Authors should report other key methods recently studied.
Suggestion: https://doi.org/10.1016/j.susmat.2021.e00298
Response 2: Thank you for your careful comment. We have added the reference in the introduction part. By citing literature, we further demonstrate the feasibility of using enzyme preparations for shrink-resistant finishing of wool.
Point 3: Authors should highlight the advantage of L-Cysteine – Protease combination over other methods.
Response 3: It is thankful for the recommendation. In order to replace chlorination shrink-proof technology, a large number of reagents and methods have been studied to treat wool fibers. Among them, protease treatment is the most promising finishing method to achieve industrialization. However, due to the strong chemical inertia of wool scale layer, the effect of protease on fibers is always slow. In order to solve the above problems, a combined solution system of L-cysteine and protease was developed in this study, and the rapid shrink-proof finishing of wool was realized by multiple padding finishing. At the same time, this method will not cause harm to the environment and human body, and has a good development prospect. Based on your suggestion, we have revised the abstract and conclusion of the manuscript to clarify the highlights of this study.
Point 4: Line 57 protease k, “K” should be capitalized.
Response 4: Thank you for your careful comment. We are very sorry for some errors in the article, and the “k” in line 57 has been revised. The revised section as followed:
Li[19] used a special protease K to biologically modify wool fibers, which can hydrolyze the scale layer effectively.
Point 5: Line 93 Kindly mention the name of the reducing agents
Response 5: Thank you for your careful comment. We have added the reductant information in the revised manuscript. The revisions are committed as followed:
The effect of L-cysteine on disulfide bond in wool was explored by comparing the properties of L-cysteine and other four reducing agents(Tris (2-carboxyethyl) phosphine hydrochloride, Sodium sulfite anhydrous, Sodium bisulfite, Sodium hydrosulfite).
Point 6: Line 99-100: Repeating the last statement of the abstract
Response 6: Thanks again for your constructive comment. According to your suggestion, we have revised the inappropriate sentence to the last statement of the abstract.
Point 7: The material and methods section should be before the results section. Please check with journal style.
Response 7: Thank you for your careful comment. We are sorry for some errors in the article. The manuscript we originally submitted was written in the format you described, and then the editor reordered the sections of the article according to the journal's format requirements. This is also the reason why some tables and formulas in the article are out of order.
Point 8: Line 345-346: Check English and grammar: Authors should say what and how they have processed their samples.
Response 8: Thanks again for your constructive view. We are very sorry for some grammar mistakes and illegibility in article. Based on your suggestion, we have rewritten the section 3.5.1 of the revised manuscript.
Point 9: Line 355: “electron microscopy” should be “Electron Microscopy”
Response 9: Thank you for your careful comment. We have modified the inappropriate word format in the revised manuscript according to your suggestion.
Point 10: Line 443: Replace “shrunk” with “shrink”
Response 10: Thanks again for your constructive comment. We are very sorry for some spelling mistakes in article. We have double-checked the English writing and the revised contents are marked in the revised manuscript.
Point 11: Equation numbers throughout the manuscript are not consistent and not in line.
Response 11: As mentioned in point 7, some formulas and tables are in the wrong order due to a change in the layout of the paper. We have modified the formula serial number in the revised manuscript.
Point 12: Please cross-check the values of the felt ball test compared with the existing literature.
Response 12: Thank you for your valuable advice. According to your comment, we have added relevant literature in Section 2.3 to compare and analyze the felt ball test results in this paper with those in other study. The revised section as followed:
Iglesias et al[22] used the biosurfactant extracted from Bacillus Subtilis for wool pretreatment, and then utilized the extracellular proteolytic extract of Bacillus to perform shrink-proof finishing on wool fibers. The felt ball density results shown that this method can significantly reduce the fiber felting tendency, and without a significant loss in wool tensile strength. The felting ball density of the treated wool in this literature is 49kg/m3, which is similar to the density value in this study. The results indicate that the felting shrinkage of fibers after finishing was effectively improved.
Special thanks to you for your helpful comments.
Reviewer 3 Report
The manuscript is well written and provides useful and reliable information about Wool Shrink-Proof Finishing Technology. The data are well described and discussed. The figures and tables well present the details of the work.
I suggest the acceptance of the manuscript with the current format.
Author Response
Response to Reviewer 3 Comments
Dear editors and reviewers:
First of all, thank you for your kindly and hard work for our paper. We are very grateful to have been given the opportunity to revise our manuscript. We have carefully considered the comments and modified the manuscript accordingly. Revised portion were marked up using the “Track Changes” function and we hope the improved manuscript will meet your magazine’s standard.
Special thanks to you for your helpful comments.
Reviewer 4 Report
Bo Li et al. in the manuscript entitled “Development of Environmentally Friendly Wool Shrink-Proof Finishing Technology Based on L-Cysteine/Protease Composite Solution System” developed and optimized a method based on a mixture of L-cysteine and protease 16L to treat wool fibers. The treatment was performed to reduce the wool scaling and consequently reduce the natural tendency of wool to felt. The authors used a response surface method to optimize the process. The study is well-written and the subject interesting. However, I do have some concern about the robustness of the proposed models.
1) I would suggest to the author to include some relevant paper on fibers optimization by RMS
(https://doi.org/10.1021/acsbiomaterials.0c01657) and on keratin (https://doi.org/10.3144/expresspolymlett.2019.10).
2) The author should add the equations of the 95% Confidence intervals (High and Low).
3) In Table 3 too many terms appear to be insignificant (P>0.05), the author should justify their presence in the proposed model. In addition, the Lack of fit of the same model is significant (P<0.05).
4) Is there any replica? Whenever RSM is used on biomaterials processing replication should be always performed to take into account the natural variability of the material (at least 3 trial for each process). This may be the reason of the lack of fit of the model of Table 3.
5) How many fibres were mechanically tested? The author should consider to test several fibers for each trial and provide a value and a standard deviation. Then the mean value should be modeled by RSM.
6) In Table 5 too many insignificant terms were present (P>0.05), the author should justify their presence in the proposed model.
7) Which is the r^2 values of the proposed models compared to the collected data?
Author Response
Response to Reviewer 4 Comments
Dear editors and reviewers:
First of all, thank you for your kindly and hard work for our paper. We are very grateful to have been given the opportunity to revise our manuscript. We have carefully considered the comments and modified the manuscript accordingly. Revised portion were marked up using the “Track Changes” function and we hope the improved manuscript will meet your magazine’s standard. The main corrections and the responds are listed as follow:
Point 1: I would suggest to the author to include some relevant paper on fibers optimization by RMS
(https://doi.org/10.1021/acsbiomaterials.0c01657) and on keratin (https://doi.org/10.3144/expresspolymlett.2019.10).
Response 1: Thank you for your valuable advice. Based on your comments, we have added the above references to the relevant section on RMS in the manuscript. The revisions are committed as followed:
(1) It can be seen from Table 2 that the P-value (0.0389) in the breaking strength regression model was less than 0.05, indicating that the model was significant. (P-value less than 0.05 means significant difference, P-value less than 0.01 means extremely significant difference.)[29]
(2) The peak found at 1680~1610cm-1 was associated with the stretching vibration of C=O (amide I).[40]
Point 2: The author should add the equations of the 95% Confidence intervals (High and Low).
Response 2: Thank you for your careful comment. According to your suggestion, we have supplemented the 95% confidence intervals for the individual response factors in Section 2.2.
Point 3: In Table 3 too many terms appear to be insignificant (P>0.05), the author should justify their presence in the proposed model. In addition, the Lack of fit of the same model is significant (P<0.05).
Response 3: It is thankful for the recommendation. The dispersion of wool fiber is large, especially the treated fiber sample. Therefore, when simulating the statistical model, more samples should be selected to optimize the variable parameters. 50 of each fiber sample were selected for strength testing in our study, but the number of samples may still be small. This may be one of the reasons for the poor fitting effect of the model, making the Lack of Fit term appear significant in the results. At the same time, this also leads to the insignificant interaction among some influencing factors (AC, AB, BC) in the model. However, since each influencing factor and its value range in the experiment were obtained through the relevant experiments in the previous period, it should have a significant effect on the wool performance. Therefore, the reasons for the lack of model fit and the insignificant effects of some factors may be mainly caused by insufficient sample selection and inappropriate data processing methods. In future research, we will focus on optimizing the response surface experiment to improve the fitting degree of the model.
Point 4: Is there any replica? Whenever RSM is used on biomaterials processing replication should be always performed to take into account the natural variability of the material (at least 3 trial for each process). This may be the reason of the lack of fit of the model of Table 3.
Response 4: Thank you for your constructive comment. We are very sorry for the shortcomings and inadequacies of the RSM experimental design. During the RMS experiment, we carried out two parallel experiments under the same conditions, and finally selected a group with better simulation results for discussion. However, we have realized that this experimental method is not helpful to the accuracy of the simulation results. In the follow-up study, we will further optimize the statistical model of wool fiber finishing process according to your guidance. In addition, we have also explained the reasons for some of the model's lack of fit in Section 2.2 of the revised manuscript.
Point 5: How many fibres were mechanically tested? The author should consider to test several fibers for each trial and provide a value and a standard deviation. Then the mean value should be modeled by RSM.
Response 5: Thank you for your careful comment. The tensile strength results for each wool sample in our study were the average of the 50 fiber results. According to your suggestion, we will increase the number of samples for fiber strength test in the subsequent research and optimize the calculation method of the average value.
Point 6: In Table 5 too many insignificant terms were present (P>0.05), the author should justify their presence in the proposed model.
Response 6: This problem is similar to point 3. Although the “Lack of Fit” term of the weight loss rate model is not significant and the R2 value is large (0.963), there are still some items that are not significant. This phenomenon may still be related to the insufficient number of test samples.
Point 7: Which is the r^2 values of the proposed models compared to the collected data?
Response 7: Thank you for your careful comment. We are very sorry for the missing important information in the RMS analysis. We have added and analyzed R2 and related test results in Tables 2 to 4 of the revised manuscript. From the test results, it can be found that the R2 values of fiber tensile strength and felt ball density are 0.839 and 0.757, indicating that the fitting effect of these two models on the experimental results is poor. In contrast, the model of fiber weight loss rate (0.962) can better fit the experimental results. The related results show that the model we established fails to solve the problem of large fiber dispersion very well. This will be the focus of our future research to improve. We also clarify the problems and shortcomings of the RMS model in the revised manuscript.
Special thanks to you for your helpful comments. Your suggestions for the manuscript will provide us with great help in deeply understanding the RMS method and optimizing the shrink-proof finishing process.
Round 2
Reviewer 1 Report
see attachment for my comments.

Author Response
Response to Reviewer 1 Comments
Dear editors and reviewers:
First of all, thank you for your kindly and hard work for our paper. We are very grateful to have been given the opportunity to revise our manuscript. We have carefully considered the comments and modified the manuscript accordingly. Revised portion were marked up using the “Track Changes” function and we hope the improved manuscript will meet your magazine’s standard. The main corrections and the responds are listed as follow:
Point 1: Line 15-16: <In this study, L-cysteine will be compounded with protease to form a combined solution system for shrink-proof finishing of wool fibers>.
“Was” should be used instead of “will be” because the work was carried out before.
Suggesting: L-cysteine was combined with protease to form a treatment solution system for shrink-resistant finishing of wool fibers.
The results only showed the reduction of felting shrinkage. Can the developed process achieve machine washable wool?
Response 1: Thank you for your constructive comment. Based on your suggestion, We are very sorry for some grammar mistakes in article. According to your suggestion, we have modified the relevant statements and double-checked the English writing in the revised manuscript.
L-cysteine/protease shrink-proof technology is mainly used to process wool tops. The finishing process parameters and fiber properties after treatment were analyzed in this study. In order to determine whether the wool can reach the machine washable standard, it is necessary to conduct washing shrinkage test on the fabrics. However, the process of spinning and weaving wool fibers into knitted fabrics is complicated and requires a large number of fiber samples, so it is difficult to evaluate the shrink-proof finishing effect quickly. In order to effectively and conveniently determine the finishing effect of wool, the felt ball density test was used to analyze the change of fiber felting properties. The following work will focus on whether the finished fibers can meet the international standards of wool machine washability after being woven into fabrics of different specifications. At the same time, in order to avoid improper expression and clarify the limitations of this study, we revised the conclusion in manuscript.
Point 2: Line 14, what does “it” mean?
Response 2: It is thankful for the recommendation. We rewrote the sentence in line 14 to ensure that readers do not get confused about the content of the article. The revisions are committed as followed:
Therefore, it is of great significance to develop an environmentally friendly and effective shrink-proof finishing technology.
Point 3: what does “L/PSCS” stands for? Does this “L/PSCS” need in the abstract?
Response 3: Thank you for your careful comment. In the original manuscript, L/PSCS was the abbreviation of "L-cysteine/protease shrink-proof combined solution", which is mainly used to make it easier for readers to understand relevant research content when reading papers. Based on your suggestion, we have modified the “L-cysteine/protease shrink-proof combined solution" to “L-cysteine/protease treatment solution system" and abbreviated it as "L/PTSS" in the revised manuscript.
Point 4: Line 21-23 in the abstract: <The results indicated that when the concentrations of L-cysteine and protease 16L were 9g/L and 1g/L respectively, wool was padding for 5 times at 50℃ and each immersion time was 30s, the shrink-proof effect of treated fibers was the most obvious>.
This sentence in the abstract doesn’t mean anything without knowing the detail of the process. If protease 16L is a commercial name, “Protease 16L” should be used. However, is it necessary to give the commercial name “Protease 16L” rather than just “protease”? same to Fig.1.
“most obvious” is very vague.
Response 4: Thank you for your valuable advice. According to your suggestion, we have added the processing method of wool fibers before this sentence in the abstract to avoid confusion to readers. At the same time, the " most obvious " in the conclusion have been rewritten.
Because there are many kinds of proteinases and their performances differ greatly, the selected proteinases will have a direct impact on the shrinkproof finishing effect. Our research team found that the protease 16L produced by Novozyme Biotechnology Co., Ltd. has excellent stability under reducing conditions, which can meet the design requirements of reducing agent/protease solution system. In addition, other researchers have also applied the Protease 16L to the scale stripping of wool, proving that it has a good reaction effect on wool. Therefore, indicating the protease species in this paper is beneficial to the generalization of the research results.
Point 5: Line 20-21 <The response surface method was used to analyze the optimal parameters of the finishing process>
What is the “response surface method” used? This analytical method could not analyse the optimal parameters directly. I reckon that this method was used for analysis of fibre surface.
Response 5: Thank you for your careful comment. The response surface methodology (RSM) is a statistical method to solve multivariable problems. This method is to obtain a certain amount of data by reasonably designing experiments. Then, multiple quadratic regression equation is used to fit the functional relationship between factors and response values. Finally, the optimal process parameters are obtained by analyzing the regression equation. The RSM can determine the approximate mathematical model through a limited number of tests to guide the subsequent experimental design. This paper also cited some articles that used RSM to analyze the optimal process of fiber material modification.
Point 6: Line 25: <the fiber scale structure was stripped evenly>
However in Line 151, it stated that “..uneven treatment”
Response 6: Thank you for your careful comment. The content in line 25 of the abstract refers to that L/PTSS has an effective and uniform peeling effect on wool scales under the optimal processing conditions. The "uneven treatment" in line 151 mainly refers to that the dispersion of fiber breaking strength is increased compared with that of raw wool after the fibers are treated with different kinds of reducing agents by immersion, which may be caused by uneven treatment effect. The fiber treatment methods analyzed in these two parts are different.
Point 7: Line 29: What are the “other methods”?
Response 7: We are very sorry for some inappropriate expression in articles. “Other methods” mainly refer to chlorination method and other protease shrink-proof technologies. We have rewritten this sentence and marked it in the revised manuscript.
Point 8: does this research work presented in the manuscript is just providing research references rather than knowledge?
Response 8: In this study, the wool top was treated with a mixed solution system composed of environment-friendly reagents and enzymes by continuous multiple-padding method. The technology can improve the felting property of wool in a short time, and will not cause excessive damage to the mechanical properties of the fibers. The research results provide an experimental and theoretical basis for further development of wool shrinkproof finishing methods that can be industrialized. The “research references" in the original paper may not be appropriate, so we have revised it to “research foundation" in the revised manuscript.
Point 9: Line 38-40, <However, it is precisely because of the unique physical structure and characteristics of wool that the products are prone to obvious felt shrinkage changes when subjected to external mechanical action in a humid and hot environment>.
This sentence doesn’t link to the previous sentence. The meaning of this sentence is not clear.
Response 9: We are sorry for the inappropriate expressions of some sentences in articles, and the related content in the introduction have been rewritten. The revisions are committed as followed:
These excellent properties are closely related to the structure and performance of wool. However, it is also precisely because of the unique scale structure and mechanical properties of fibers that the wool products are prone to obvious felt shrinkage changes when subjected to external mechanical action in a humid and hot environment.
Point 10: <Aiming at the directional friction effect (D.E.F.)……..>
“the directional friction effect (D.E.F.)” wasn’t mentioned before this sentence. “Aiming to alter (modify) the directional friction effect (D.E.F.)”?
Response 10: Thank you for your careful comment. According to your suggestion, we have rewritten this sentence in the revised manuscript.
Point 11: Line 62-63: what is the special for “protease K”? What are three-functions of the protease? Does the protease have the function of reducibility?
Response 11: According to the literature, proteinase K is a keratinolytic protease with broad specificity. This protease preferentially cleaves peptide and ester bonds at the C terminal of aromatic amino acids, sulfur-containing amino acids and hydrophobic amino acids. Therefore, proteinase K was used by the authors to improve the shrink-proof, anti-pilling and dyeing properties of wool fabrics.
In another reference, the authors covalently bonded the protease molecule with with poly (ethylene glycol) bis (carboxymethyl) ether (HOOC-PEG-COOH) and L-cysteine to develop a novel “trifunctional protease” with reducibility, hydrolysis and localization. The experiment proved that the trifunctional protease can remove the scale layer on wool surface under relatively mild conditions, so as to achieve the effect of shrink-proof finishing.
We have modified the content of relevant references in the revised manuscript to more accurately introduce the work of other researchers.
Point 12: Line 69: what is “16.37%”? Can the research methods from the literature satisfy the demand for wool machine washability? Are they just complex and time-consuming?
Response 12: Thank you for your valuable advice. According to the International Wool Textile Organization (IWTO) test method TM 31, if the area shrinkage of wool fabric with specific structure is less than 8% after washing shrinkage test, it means that the wool sample has reached the machine washable standard. Therefore, the conclusion of the cited literature shows that the finishing method can satisfy the requirements of wool machine washability.
It has been proved that selecting appropriate protease to treat wool fibers can effectively meet the requirements of shrink-proof finishing. However, due to the slow hydrolysis ratio of most proteases on wool scales, other reagents or methods should be combined to improve the efficiency. Therefore, the existing protease shrink-proof technologies generally have the problems of complex processes or time-consuming, leading to their inability to be industrialized.
Point 13: Line 116: omit “reaction”
Response 13: According to your suggestion, we have modified the relevant contents in the revised manuscript.
Point 14: Line 120, “Tris (2-carboxyethyl) phosphine hydrochloride” should be moved to the front of TCEP in Line 120 rather than appearing in line 122.
Response 14: Thank you for your valuable advice. According to your suggestion, we have modified the relevant contents in the revised manuscript.
Point 15: Line 147- 149. How “the change range of TCEP treated samples (24.1%) was the largest”. 24.1% is lowest even at the outside of the range between 49.5% and 30%. Confusing here.
Response 15: Thank you for your valuable advice. We are very sorry for the inappropriate expressions of some sentences in articles. The meaning expressed in the manuscript is that the elongation at break of wool fibers changes obviously after treatment with reducing agent. In the test results, the elongation at break of raw wool was 49.5%, and it was reduced to about 30% after treatment. The result of TCEP treated fibers was 24.1%. Compared with the samples treated with other reducing agents, it has the most changes. In order to avoid misunderstanding to the readers, we rewrote the relevant content in the revised manuscript.
Point 16: Line 163 <….by multiple linear regression.>. however, the equation 4 is not linear regression.
Table 2, soaking time and padding times are still confusing.
Response 16: Thank you for your valuable advice. We are very sorry for some errors and illegibility in the article. In the revised manuscript, we have changed the "multiple linear regression" to "multiple regression function", and replaced the "Padding times" with "The times of padding".
Point 17: Line 294: what does “the number of specific amino acids”? obvious this paragraph discusses the concentration of proteins or polypeptides in the residual bath or treatment solutions. It is confusing with the number of amino acids which was related to the research method described in 3.5.11 for amino acid contents of fibres.
Response 17: Thanks again for your constructive comment. According to your suggestion, we have changed "the number of specific amino acids" to "The content change of specific amino acids" in the revised manuscript.
Point 18: In the results and discussion, WF and SP-WF should not be used.
Response 18: Thank you for your valuable advice. We have revised "WF" and "SP-WF" to "raw wool" and "treated fibers" respectively in the revised manuscript.
Point 19: Line 358, authors should provide more information of wool fibres used. 70s wool fibres?
Response 19: Thank you for your valuable advice. According to your suggestion, we have given a more detailed description of the wool used in the study.
Point 20: Equation 5 should be used for the weight loss (%) rather than weight loss ratio (%). According to the equation for calculation, the results of weight loss (%) for treated samples should be minus (-), but the results are all positive (+).
Response 20: Thank you for your careful comment. We are very sorry for the errors in the article. Formula 5 in the original manuscript is wrong, and the correct form should be "(M0-M)/M0×100". In addition, due to the large dispersion of wool fibers, "weight loss ratio" can more accurately reflect the effect of finishing process on fibers. The "weight loss" results require that the raw wool samples used each time have a relatively consistent number and morphology, so as to ensure the comparability of different test results. Therefore, it is more appropriate to use "weight loss ratio" in this paper.
Point 21: L427-428: how could be “dye cup”? are they pots or beakers?
Any rotation and metal balls used in the felt ball density test?
Response 21: It is thankful for the recommendation. In the felt ball density analysis, the infrared sample dyeing machine we used is similar to the equipment structure in the following figure. During the experiment, the fiber samples and the finishing solution were stored in the dye pots for constant temperature finishing. It is unnecessary to use rotation or metal balls in the treatment process, and the fibers put into the dye pots after the treatment will form a ball, so this experiment is called felt ball density test.
With your reminder, we realized that it is not appropriate to use "dye cup" here, so we revised it to "dye pots" in the manuscript.
Point 22: Line 436 mistake q.
Response 22: Thank you for your careful comment. We are sorry for the spelling mistakes in article. According to your suggestion, we have modified the letter in the revised manuscript.
Point 23: Conclusion Line 495-500 <Compared with other shrink-proof methods, the L/PSCS technology developed in this study solved the problem of slow action rate of 496 protease on wool while achieving the finishing effect by reasonably designing the finishing solution system and processing method. At the same time, the reagents used in this method are environment-friendly materials, which meet the requirements of industry development and have broad development prospects.>
This is not clear at all.
Response 23: Thank you for your valuable advice. According to your suggestions, we have modified some contents in the conclusion to highlight the limitations of this study and the focus of subsequent research. The revisions are committed as followed:
Compared with other protease shrink-proof methods, the L/PTSS technology developed in this study solved the problem of slow action rate of protease on wool by reasonably designing the finishing solution system and processing method. After treatment, the shrink-proof property of the wool fiber was significantly improved, and its mechanical properties were also not excessively damaged. At the same time, the reagents used in this method are environment-friendly materials, which has broad development prospects and research value. The follow-up work will further optimize the L/PTSS finishing process to ensure that the wool of different qualities can reach the industry standard of machine washable after treatment, and gradually promote the industrial application of this technology.
Point 24: Authors should be careful for claiming the successful shrink-proof finishing. There are still doubt about the tensile strength of individual fibres because the difference of wool fibres in their fibre diameters and their deviation of tensile test results are so big to conclude the difference. It is not clear whether the treated wool fibres are machine washable with acceptable damage.
Response 24: In the revised manuscript, the achievements of this paper have been modified, so as to clarify the limitations of the existing research. The specific revised content is shown in point 23.
Point 25: Authors used “combined solution system” in the revised manuscript. The “combined” could mislead.
Response 25: It is thankful for the recommendation. We have changed "combined solution system" to "treatment solution system" in the article. At the same time, we also replace the "combined solution" with "mixed solution" in other contents.
Special thanks to you for your helpful comments.

Reviewer 4 Report
The authors replied to all my concerns, the paper should be accepted for publications.
Author Response
Response to Reviewer 4 Comments
Dear editors and reviewers:
First of all, thank you for your kindly and hard work for our paper. We are very grateful to have been given the opportunity to revise our manuscript. We have carefully considered the comments and modified the manuscript accordingly. Revised portion were marked up using the “Track Changes” function and we hope the improved manuscript will meet your magazine’s standard.
Special thanks to you for your helpful comments.